# Photosynthesis, Biomass Production, Nutritional Quality, and Flavor-Related Phytochemical Properties of Hydroponic-Grown Arugula (*Eruca sativa* Mill.) 'Standard' under Different Electrical Conductivities of Nutrient Solution

**Teng Yang [1], Uttara Samarakoon [1,*], James Altland [2] and Peter Ling [3]**

[1] Agricultural Technical Institute, The Ohio State University, Wooster, OH 44691, USA; yang.5542@osu.edu
[2] United States Department of Agriculture Agricultural Research Service, Wooster, OH 44691, USA; james.altland@usda.gov
[3] Department of Food, Agricultural and Biological Engineering, The Ohio State University, Wooster, OH 44691, USA; ling.23@osu.edu
* Correspondence: samarakoon.2@osu.edu; Tel.: +1-330-287-1241

**Abstract:** Arugula (*Eruca sativa*) is cultivated using hydroponic techniques in greenhouses to fulfill high year-round demand, but its nutrient management in hydroponic production has not yet been standardized, potentially leading to limited quality and productivity. Aiming to address this issue, we investigated the effect of electrical conductivity (EC) on yield, nutritional and phytochemical properties of arugula. The model cultivar arugula 'Standard' was grown at four different EC levels (1.2, 1.5, 1.8, and 2.1 dS·m$^{-1}$). Our results indicated photosynthetic properties, SPAD, leaf area, yield and dry weight increased with increasing EC from 1.2 to 1.8 dS·m$^{-1}$. Foliar nutrient content increased with higher EC, but nutrient solution with 2.1 dS·m$^{-1}$ showed a significant decline in N, Ca and most of the micronutrients including Fe, Zn, Mo, Cu, B and Mn. Total glucosinolates, total chlorophyll and total carotenoids concentrations increased with increasing EC. In addition, total anthocyanin content was highest in plants grown in EC 1.2 and 2.1 dS·m$^{-1}$, demonstrating a stress response when grown in extreme EC levels. Our results further indicated a rapid accumulation of nitrate with higher EC, potentially detrimental to human health. This research demonstrated the optimal EC range would be 1.5 to 1.8 dS·m$^{-1}$ for arugula in hydroponic production systems based on yield, quality criteria and human health considerations.

**Keywords:** *Eruca sativa*; hydroponics; mineral uptake; nitrates; total anthocyanin; total chlorophyll; total carotenoids; total glucosinolates

## 1. Introduction

Arugula or rocket (*Eruca sativa*) is a leafy vegetable in the family *Brassicaceae*. It has long been used as an ingredient in the cuisines of many countries such as Italy, Morocco, Portugal and Turkey and has been a popular culinary ingredient in the United States since the 1990's [1]. Arugula is a fast-growing (usually 20–30 days after germination) and cool-season crop [2] that can be recut and harvested continuously. Arugula is cultivated in hydroponics and greenhouses to provide higher quality and greater yields to fulfill year-round demand [3,4]. It is commonly used as food for its pungent or bitter flavor and abundant nutrients (potassium, sulfur, iron, and vitamins A and C) in their edible leaves [5]. Arugula has also been used for many medicinal purposes [2] as its leaves contain large quantities of health-promoting compounds, mainly contributed from glucosinolates and antioxidants with proven pharmaceutical and anti-cancer properties [6,7]. Despite the numerous studies on utilization of arugula, scientific studies on cultural practices affecting its commercial production are still lacking, only information available is from anecdotal experience of producers [8]. Even more, many growers simply follow the recommended

practices for lettuce (*Lactuca sativa*) in arugula production, potentially limiting quality and productivity. Therefore, there is a need to evaluate and optimize nutrient management of arugula in hydroponic and other systems to provide maximum quality, productivity, and profit.

For commercial vegetable production, developing standardized culture techniques result in greater production efficiency is desirable [9]. There is an expansion in the vegetable industry to switch from open field or traditional soil culture to controlled environment and soilless cultivation systems, frequently referred to as hydroponic production [3]. Because hydroponics provides better control of plant growth, it is possible to achieve product quality and productivity through careful management of dissolved oxygen concentration, temperature, nutrient composition, and pH and electrical conductivity (EC) of the nutrient solution [10].

In hydroponic production systems, EC management is one of the most important and manageable cultural practices that affects the visual, nutritional, and phytochemical quality of leafy vegetables. In hydroponic nutrient solutions, essential nutrient elements are dissolved in appropriate concentrations and relative ratios to achieve the normal growth of plants. In practice, real time monitoring of individual nutrient elements has both economic and technical limitations [11]; instead, tools for managing EC are simple and robust, allowing their use on farms with little training or scientific background. Therefore, it is of interest to identify EC levels that optimize arugula growth and yield in hydroponic production systems.

Arugula is considered moderately sensitive to nutrient environment [12], its nutrient management could be specifically developed aiming to enhance the health-promoting phytochemicals (plant secondary metabolites) such as glucosinolates and antioxidants, while paying close attention to improve the sensorial traits [13,14]. Glucosinolates are particularly abundant in the *Brassicaceae* family which impart the pungent and bitter taste of arugula [15–17]. Some studies have shown a relationship between the glucosinolates and the plant species as well as the environmental, nutritional and growth conditions [18]. However, little evidence is available on the effects of nutrition levels in hydroponic production on glucosinolates content. In addition, antioxidants are also a group of plant secondary metabolites which are abundant in arugula leaves in response to osmotic stresses such as salinity, light, pH, and temperature [2,19]. Antioxidants include flavonoids, carotenoids, pigments, etc. Anthocyanin is one of the most important flavonoids which also could affect the color of arugula leaves. Consequently, it is critical to investigate the nutrient uptake and phytochemical accumulation (especially glucosinolates, antioxidants, carotenoids and chlorophyll) in hydroponic grown arugula under different EC levels of fertilizer solution in order to optimize its nutrient management.

Nitrate accumulation in horticultural crops and its negative effects on cardiovascular system and cancer incidence is well documented [2,20], which is a big concern in Europe [2] and could become important in US. Although there may be a range of beneficial vascular effects, these remain unproven [20]. Numerous researches also indicated the high correlation between root-zone nitrogen concentration and crop nitrate accumulation in both conventional [21] and controlled [22] environments. In addition, nutrients in hydroponic systems are highly available as compared to conventional production, which lead to higher potential of accumulation. However, while arugula is a hyper-accumulator of nitrates [23,24], there was still limited investigation aiming to optimize nutrient management in hydroponic-grown arugula to reduce its nitrate accumulation.

Thus, with the aim of determining the optimum EC levels for high quality hydroponic-grown arugula, the present research evaluated the EC effect on yield, physiological and nutritional proprieties of arugula cultivar 'Standard' grown with a nutrient film technique (NFT) hydroponic system.

## 2. Materials and Methods

### 2.1. Plant Material and Growing Conditions

The experiments were conducted from December 2019 through January 2020 using a nutrient film technique (NFT) system (CropKing, Lodi, OH, USA) established in a double polyethylene-plastic covered greenhouse at the Ohio State University, Wooster Campus, Ohio (40.78° N, 81.93° W). Four different nutrient solutions representing for EC-treatments were stored in four ultra-violet stabilized plastic tanks separately. Each tank was randomly connected to four of 16 growing channels that were each 4 m long (Figure 1). Two border channels were connected to a separate tank. Each growing channel was 0.2 m apart with the capacity to grow 18 plants. Each plant grown in the channel was 0.2 m apart with each other. A galvanized steel frame was used for supporting all the growing channels. Each reservoir tank was equipped with one high-efficiency circulation pump (Model 3WY90; Dayton Electric Mfg., Niles, IL, USA) to deliver nutrient solution to the growing channel and drain the nutrient back to the reservoir tank.

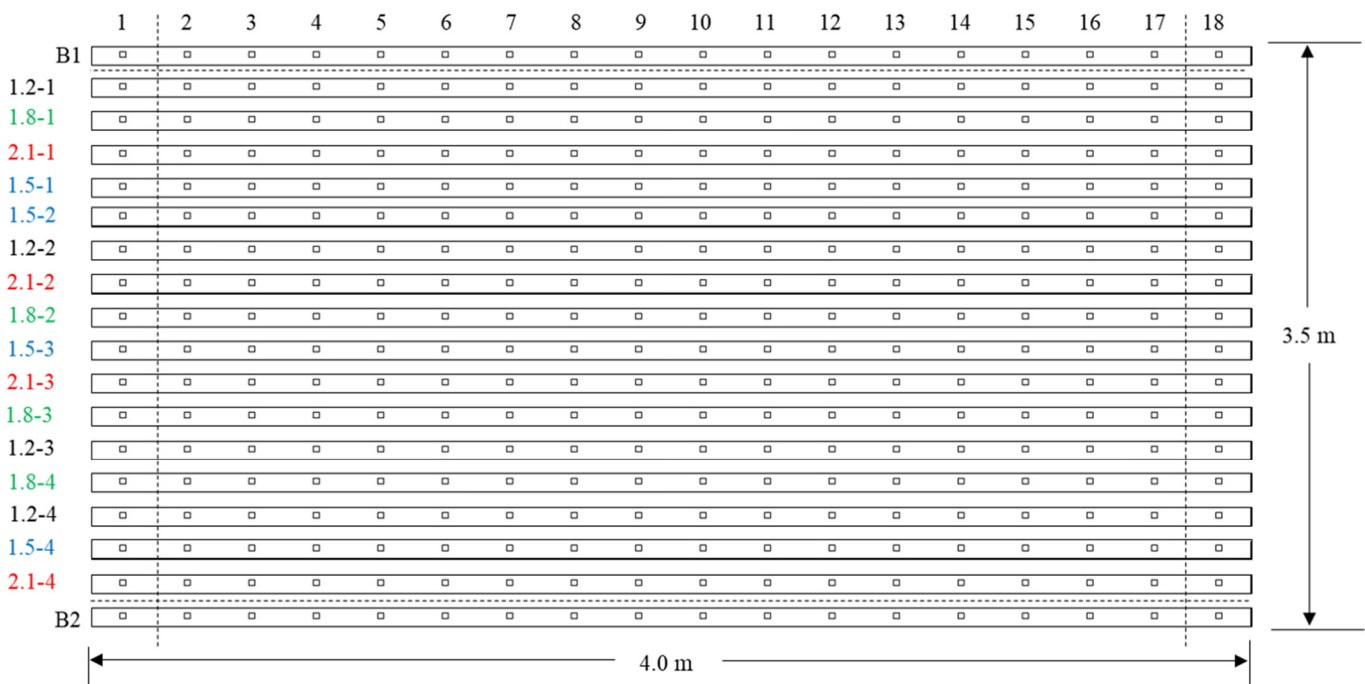

**Figure 1.** Growing channel layout showing the location of treatments replications.

The air temperature and air humidity of the greenhouse were measured every 10 s with a humidity and temperature probe (INTERCAP® HMP50; Vaisala, Helsinki, Finland). Photosynthetic photon flux density (PPFD) was provided by natural light and high-intensity discharge (HID) lamps (400-Watt high-pressure sodium (HPS), Energy Technics Horticulture Lighting, York, PA, USA) for 16 h per day and measured every 10 s with a sun calibration quantum sensor (SQ-110-SS; Apogee Instruments, Logan, UT, USA). Air temperature, air humidity and average light intensity were logged every 10 s with a micrologger (CR3000; Campbell Scientific, Logan, UT, USA). During the experiment, the average (±standard error) day and night air temperature, air humidity and daily light integral of PPFD were $18.51 \pm 0.05$ °C, $16.76 \pm 0.02$ °C, $50.85 \pm 0.14$ % and $17.03 \pm 1.71$ mol·m$^{-2}$·d$^{-1}$, respectively.

The experiment was conducted using Arugula (*Eruca sativa*) 'Standard' (Johnny's Selected Seeds, Albion, ME, USA) as the model plant. A water-soluble fertilizer (Hydro Grow Leafy Green Fertilizer; 4.3% N-9.3% P-35% K: Crop King, Lodi, OH, USA) at 100 mL·L$^{-1}$ and calcium nitrate (CropKing) at 78 mL·L$^{-1}$ was used as fertilizer stock solution which were used to prepare feeding solutions with four different EC treatments in the study.

Seed germination was done in 162-cell foam (Horticubes®; Smithers Oasis, Canton, OH, USA) using a propagation system for hydroponic crop production with nutrient circulation (average temperature at 18.3 ± 0.02 °C). A diluted solution with EC 1 to 1.2 dS·m$^{-1}$ and pH 5.8 was used as first leaves appeared as per the recommendation. Arugula seedlings were transplanted at 3 weeks from germination with an average plant height of 5.3 ± 0.3 cm, leaf number of 3.0 ± 0.1, leaf area of 5.62 ± 0.36 cm$^2$, SPAD (an index of chlorophyll content per unit leaf area) of 24.12 ± 1.17, fresh weight of 0.59 ± 0.03 g, and dry weight of 0.05 ± 0.003 g. The time from transplanting to harvest was 4 weeks when plants became fully mature.

## 2.2. Treatments and Experimental Design

There were four EC treatments investigated in this experiment: 1.2, 1.5, 1.8, 2.1 dS·m$^{-1}$ (Table 1). Each reservoir tank represented one EC treatment and EC was adjusted daily within ± 0.05 range using the fertilizer stock solution. After the adjustment of EC, the pH of all the reservoir tanks was adjusted to 5.82 ± 0.05 daily using 20% citric acid (reduce pH) or water (increase pH). Supplemental water (EC 0.6 dS·m$^{-1}$, pH 7.0 and alkalinity 53.9 mg·L$^{-1}$) was added into the reservoir tanks to maintain the water level, then EC and pH were adjusted according to the treatment settings. Supplemental water was collected for measurements of EC, pH, redox, and alkalinity ($CaCO_3$) using a titrator (T7; Mettler Toledo, Columbus, OH, USA) with an autosampler (InMotion Max; Mettler Toledo) and a pH probe (DGi115-SC; Mettler Toledo). The nutrient composition of supplemental water was determined with ion chromatography systems (IC 600; Thermo Fisher Scientific, Waltham, MA, USA). The total nitrogen (TN) and total organic carbon (TOC) contents of supplemental water was determined using the total organic carbon analyzer (TOC-LCSN, Shimadzu, Kyoto, Japan). EC and pH of each treatment were monitored using EC meter (COM-100, HM Digital Inc., Redondo Beach, CA, USA) and pH meter (PH-200, HM Digital Inc., Redondo Beach, CA, USA), respectively. Each growing channel had 18 arugula seedlings. Data were not collected from the plants in the edge growing channels and the two plants cultured on the edge of each growing channel to avoid the edge effects. Thus, there were 4 replicate channels for each treatment and 16 plants in each channel. Treatments were assigned to each channel in a completely randomized design.

**Table 1.** Electrical conductivity (EC) and pH of nutrient solutions used in the hydroponic production of arugula taken before (Pre-) and after (Post-) daily adjustment of the nutrient solution. Data represents the mean and standard error of 28 measurements taken daily.

| Treatments | Pre-EC | Post-EC | Pre-pH | Post-pH |
|---|---|---|---|---|
| EC 1.2 | 1.24 ± 0.07 [d] | 1.25 ± 0.01 [d] | 7.20 ± 0.08 [a] | 5.80 ± 0.01 [a] |
| EC 1.5 | 1.49 ± 0.04 [c] | 1.53 ± 0.02 [c] | 7.08 ± 0.07 [a] | 5.81 ± 0.01 [a] |
| EC 1.8 | 1.82 ± 0.05 [b] | 1.81 ± 0.01 [b] | 6.98 ± 0.09 [a] | 5.80 ± 0.01 [a] |
| EC 2.1 | 2.10 ± 0.04 [a] | 2.11 ± 0.01 [a] | 6.98 ± 0.09 [a] | 5.81 ± 0.01 [a] |

Means in columns with different letters are significantly different according to Tukey's test ($p$ = 0.05).

## 2.3. Measurement of Gas Exchange Properties and Light Response Curve

Gas-exchange and light response curves measurements were performed using a portable gas exchange system (LI-6400XT; LICOR Biosciences, Lincoln, NE) equipped with a 6-cm$^2$ leaf chamber with built-in LEDs (470 and 665-nm peak wavelengths for blue and red LEDs, respectively). Illumination was supplied at a PPF of 1000 μmol·m$^{-2}$·s$^{-1}$ by red and blue LEDs at a ratio of 9:1 under 20 °C in the leaf chamber when supplemental lighting was in use. The reference $CO_2$ concentration and flow rate through the chamber were 400 μmol·mol$^{-1}$ and 500 μmol·s$^{-1}$, respectively.

Four plants of each channel (16 plants for each treatment) were selected for the photosynthetic property measurement on the day before transplant, 16, 22, and 28 days after transplant. The third fully expended youngest leaf was selected from each plant for the measurements. The measurements of photosynthetic rate (*Pn*), stomatal conductance

($gs$), transpiration rate ($E$), and internal $CO_2$ ($Ci$) were conducted between 9:00 am and 16:00 pm at a PPFD of 1000 $\mu$mol·m$^{-2}$·s$^{-1}$. Readings were taken when the coefficient of variation (i.e., sample $CO_2$, sample $H_2O$, and flow rate) was less than or equal to 0.2% (stable), which typically occurred within 10 min. The intrinsic water use efficiency (WUE) was calculated by dividing *Pn* by *E* [25].

Light curves were performed with gradually increasing irradiance in nine steps with 5 min intervals. For each step, the irradiance was 0, 50, 75, 125, 250, 500, 750, 1000 and 1500 $\mu$mol photons·m$^{-2}$·s$^{-1}$, respectively. At each irradiance, the measurements were taken when the photosynthesis rate reached steady state (after about 10 min). Leaf chamber temperature and $CO_2$ concentration during the measurement were maintained at 20 °C and 1000 mmol·mol$^{-1}$, respectively. Then the *Pn* curves were drawn, maximum gross photosynthetic rate ($P_{gmax}$) and maximum net photosynthetic rate ($P_{max}$), light saturation point (LSP), light compensation point (LCP), respiratory rate (Rd), and apparent quantum yield (AQE) were estimated based on fitting a hyperbolic tangent based model following [26,27] as follows:

$$P_N = P_{gmax} \times \tanh\left(\frac{\varnothing_{(I_0)} \times I}{P_{gmax}}\right) - R_D \tag{1}$$

where: $P_{gmax}$—the asymptotic estimate of the maximum gross photosynthetic rate [$\mu$mol($CO_2$)·m$^{-2}$·s$^{-1}$]; *I*—the photosynthetic photon flux density [$\mu$mol (photon)·m$^{-2}$·s$^{-1}$]; $\phi(I_0)$—the quantum yield at $I_0$ = [$\mu$mol($CO_2$)·$\mu$mol (photon) $^{-1}$]; $R_D$—the dark respiration rate [$\mu$mol($CO_2$) ·m$^{-2}$·s$^{-1}$].

## 2.4. Measurement of Leaf Area and Relative Chlorophyll Content (SPAD)

Four representative plant samples were selected from each growing channel (eight plants for each nutrient reservoir tank and 16 plants for each treatment) for leaf area measurement on the day before transplant and 28 d after transplant. Each leaf of each plant sample was scanned for leaf area by using a portable laser leaf area meter (CI-202, CID Bio-Science, Inc., Camas, WA, USA), and recorded for the calculation of total plant leaf area.

On the day before transplant, 16, 22 and 27 d after transplant, four representative plant samples were selected from each growing channel (eight plants for each nutrient reservoir tank and 16 plants for each treatment) for relative chlorophyll content (SPAD; an index of chlorophyll content per unit leaf area) measurement. The SPAD readings were taken on each fully expanded leaf with a chlorophyll meter (SPAD-502, Minolta Corporation, Ltd., Osaka, Japan). Five readings per leaf were taken at the central point of a leaf between the midrib and the leaf margin and the values were averaged and recorded.

## 2.5. Harvesting and Yield Measurements

On 28 d after transplant, all plants were harvested. Plants used for the photosynthetic property measurement and leaf area measurement were not included in the final harvest due to possible mechanical disruption of tissues. During the harvest, nutrient deficiency symptoms of each plant were identified based on observation of leaf chlorosis.

Eight plant samples per channel were divided into roots and shoots and weighed for fresh weight. Then plant samples were oven-dried at 68 °C until a constant weight was reached before taking the dry weight. All dried samples were filtered through a 10-mesh screen after grinding with a sample mill (Cyclotec™ 1093, FOSS Analytical, Hillerød, Denmark) and kept in plastic vials for tissue nutrient analysis.

## 2.6. Measurement of Tissue Nutrient Analysis

Plant tissue nutrient analysis was conducted using samples of the shoot of 'Standard' arugula (6 plants per treatment) at Ohio State University's Service, Testing, and Research (STAR) laboratory (Wooster, OH, USA) to investigate the variations in nutrient uptake under different EC levels. Total concentrations of plant-essential elements (P, K, Ca, Mg, S, Al, B, Cu, Fe, Mn, Mo, Na, and Zn) were determined by microwave digestion with

HNO3 followed by inductively coupled plasma (ICP) emission spectrometry according to Jones [28]. Nitrate nitrogen in plant tissue samples were determined by the $NO_3$-N cadmium reduction method [29]. Total nitrogen in plant tissue samples were determined by the Dumas method according to Association of Official Analytical Chemists [30].

### 2.7. Measurement of Total Glucosinolates

Total glucosinolates were estimated by spectrophotometric method of Mawlong et al. [31]. A fresh plant shoot sample of 10 g was homogenized in a 2 mL vial with 80% methanol. This homogenate was centrifuged at 3000 rpm for 4 min after keeping overnight at room temperature. The supernatant was collected after centrifugation and made up to 2 mL with 80% methanol. The extraction of 100 µL was used for estimation. A volume of 0.3 mL double distilled water and 3 mL of 2 mM sodium tetrachloropalladate (58.8 mg Sodium tetrachloropalladate + 170 µL concentrated HCl + 100 mL double distilled water) were added to the sample. After incubation at room temperature for 1 h, absorbance was measured at 425 nm using a UV-Vis spectrophotometer (Genesys 180, Thermo Fisher Scientific Inc., Waltham, MA, USA). A blank was set following the same procedure was run without plant extract. Total glucosinolates were calculated by putting the absorbance value of each sample taken at 425 nm into the predicted formula

$$y = 1.40 + 118.86 \times A_{425}. \tag{2}$$

### 2.8. Measurement of Total Anthocyanins

Total anthocyanins were estimated by a modified spectrophotometric method of Sukwattanasinit et al. [32]. Dried and ground plants samples weighing 160 mg were macerated with 0.1% HCl in 75% MeOH (25 mL) at room temperature in a dark room for 24 h, and then filtered. Two 0.5 mL of sample extracts were separately mixed with 2.5 mL of KCl buffer (0.025 M, pH 1.0), and 2.5 mL of sodium acetate buffer (0.4 M, pH 4.5). After 30 min, absorbance of each mixture solution was measured at 520 nm (A520) and 700 mn (A700) using a UV-Vis spectrophotometer (Genesys 180, Thermo Fisher Scientific Inc., Waltham, MA, USA). The absorbance of the measured solution (A) was calculated by the following equation:

$$A = (A_{520} - A_{700})_{pH1.0} - (A_{520} - A_{700})_{pH4.5} \tag{3}$$

The anthocyanin content in the sample was calculated against the simulated calibration curve of methyl orange and expressed as percentage of anthocyanin content based on dried plant weight.

### 2.9. Measurement of Chlorophyll and Carotenoids

Chlorophyll a, chlorophyll b and carotenoids were extracted from 25 mg fresh leaf tissues using 100% methanol as the solvent. Samples were kept in a dark room at 4 °C for 24 h. Quantitative determination of total chlorophyll was carried out immediately after extraction. Absorbance readings were measured at 661.6 and 644.8 nm for chlorophyll pigments and 470 nm for total carotenoids. Chlorophyll and carotenoids concentrations were calculated by Lichtenthaler's formula [33]:

$$Chl_a = 11.25A_{661.6} - 2.04A_{644.8} \tag{4}$$

$$Chl_b = 20.13A_{644.8} - 4.19A_{661.6} \tag{5}$$

$$Chl_{a+b} = 7.05A_{661.6} + 18.09A_{644.8} \tag{6}$$

$$Car_{x+c} = \frac{(11.24A_{470} - 1.90Chl_a - 63.14Chl_b)}{214} \tag{7}$$

*2.10. Statistical Analysis*

Data were analyzed using JMP® for Windows, Version 14.0 Pro (SAS Institute Inc., Cary, NC, USA). Statistical differences were determined using a one-way analysis of variance (ANOVA) followed by Tukey's honestly significant difference (HSD) test ($p$ = 0.05). In addition, multiple regression analysis with simple linear and non-linear models were conducted, and regression plots were presented in the figures to display the effect of EC on productivity and plant response parameters. The significance of regression coefficient was indicated with *, ** or *** at $p$ of 0.05, 0.01 or 0.001, respectively.

### 3. Results

*3.1. Plant Growth and Yields*

Leaf area of Arugula increased quadratically with increasing EC at DAT (days after transplant) 28 (Figure 2). Those treated with EC 1.8 or EC 2.1 dS·m$^{-1}$ had higher leaf area than those treated with EC 1.2 dS·m$^{-1}$. While the data on this date were best fit with a quadratic function, leaf area only increased over the range of EC values with no observable decrease.

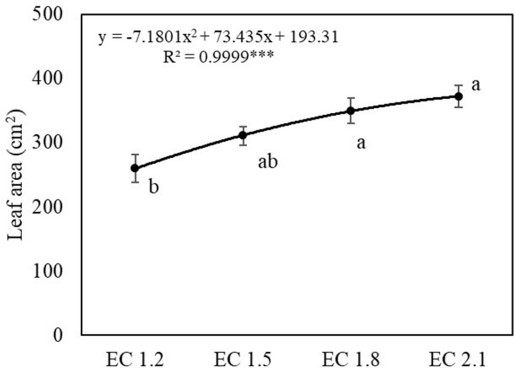

**Figure 2.** Effect of electrical conductivity (EC; 1.2, 1.5, 1.8, or 2.1 dS·m$^{-1}$) of the nutrient solution on leaf area of arugula measured 28 days after transplanting into a nutrient-film technique hydroponic system. Data points with different letters are significantly different according to Tukey's test ($\alpha$ = 0.05). Error bars represent the standard errors ($n$ = 16). *** indicates significance at $p$ of 0.001.

Leaf SPAD values increased over time for all treatments (Figure 3). By day 22, SPAD value of EC 1.2 dS·m$^{-1}$ was significantly lower than other treatments. By day 28, SPAD of plants treated with EC 1.5 and EC 1.8 dS·m$^{-1}$ were higher than those with EC 1.2 dS·m$^{-1}$.

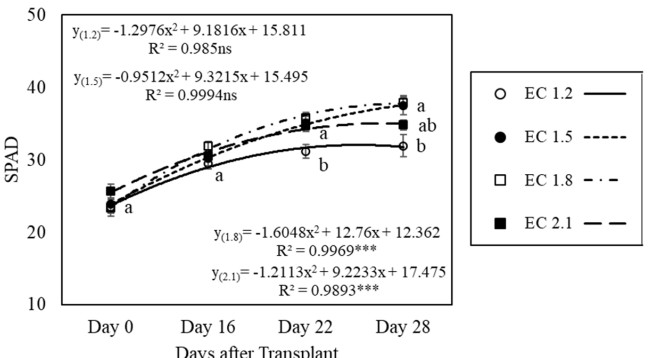

**Figure 3.** Effect of electrical conductivity (EC) of nutrient solution on relative chlorophyll content (SPAD) value of arugula on the day before transplant, and 16th, 22nd, and 28th days after transplanting into nutrient-film technique hydroponic system containing nutrient solutions with EC 1.2, 1.5, 1.8, or 2.1 dS·m$^{-1}$. Data points with different letters are significantly different according to Tukey's test ($\alpha$ = 0.05). Error bars represent the standard errors ($n$ = 16). ns or *** indicate no significance or significance at $p$ of 0.001.

Plants treated with EC 1.2 dS·m$^{-1}$ had the lowest total fresh and dry yields which were 29.67 g and 3.64 g, respectively (Figure 4). Shoot fresh yield, shoot dry yield and shoot to root ratio of EC 1.5 and 1.8 dS·m$^{-1}$ were 136% to 139%, 55% to 56% and 48% to 56% higher than EC 1.2 dS·m$^{-1}$, respectively, which indicated the higher marketable yields and aligns with the net assimilation results. Although there was no significant difference in shoot or root dry weight when EC was higher than 1.2 dS·m$^{-1}$, shoot fresh wight tended to decrease at the highest EC of 2.1 dS·m$^{-1}$, which suggest arugula maybe sensitive to high salt concentration. At the end of the study, there were plants with leaf chlorosis in EC 1.2 and 2.1 dS·m$^{-1}$ which indicated a nutrient disorder. Although yields of EC 2.1 dS·m$^{-1}$ were as high as EC 1.5 and EC 1.8 dS·m$^{-1}$, the occurrence of the yellow leaf symptoms was significantly higher than EC 1.5 and 1.8 dS·m$^{-1}$ and as severe as EC 1.2 dS·m$^{-1}$ (Figure 5). This suggests the chlorosis symptoms could have been caused by nutrient deficiency or nutrient excess causing toxicity to plants.

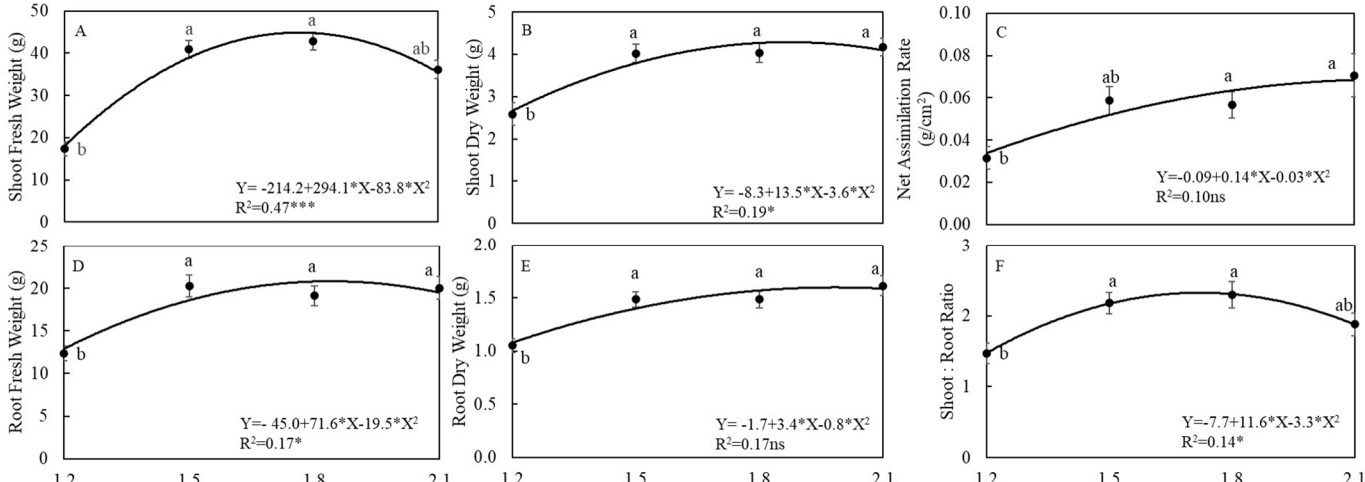

**Figure 4.** (**A**) Shoot fresh weight, (**B**) shoot dry weight, (**C**) net assimilation rate, (**D**) root fresh weight, (**E**) root dry weight, and (**F**) shoot to root weight ratio of arugula 4 weeks after transplanting into a nutrient-film technique hydroponic system containing nutrient solutions with electrical conductivity (EC) of 1.2, 1.5, 1.8, or 2.1 dS·m$^{-1}$. Data points with different letters are significantly different according to Tukey's test ($\alpha$ = 0.05). Error bars represent standard errors ($n$ = 32). ns, * or *** indicate no significance or significance at $p$ of 0.05 or 0.001, respectively.

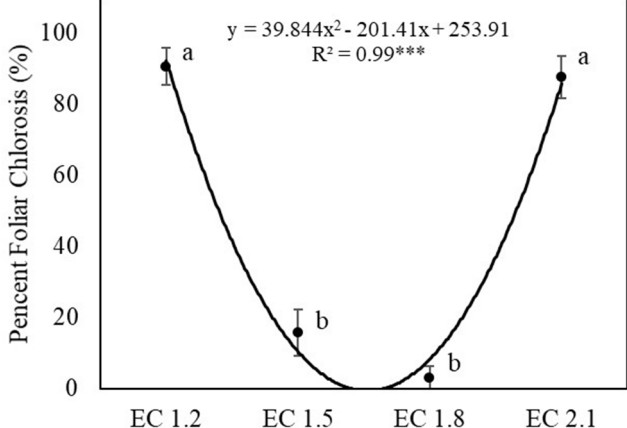

**Figure 5.** Effect of electrical conductivity (EC) of nutrient solution on a subjective rating of foliar chlorosis of arugula 4 weeks after transplanting into a nutrient-film technique hydroponic system containing nutrient solutions with EC 1.2, 1.5, 1.8, or 2.1 dS·m$^{-1}$. Data points with different letters are significantly different according to Tukey's test ($\alpha$ = 0.05). Error bars represent standard errors ($n$ = 32). *** indicates significance at $p$ of 0.001.

### 3.2. Leaf Gas Exchange Indices

Since EC of the nutrient solution is directly related to the photosynthetic metabolism [34], the leaf gas exchange characteristics were measured every week (Figure 6). Net photosynthetic rate (*Pn*, Figure 6A), transpiration rate (*E*, Figure 6B), water use efficiency (WUE, Figure 6C), stomata conductance (*gs*, Figure 6D), and intracellular $CO_2$ concentration (*Ci*, Figure 6E) changed over time, which increased in the rapid growth stage (second to third week) and decreased in the mature stage (last or fourth week).

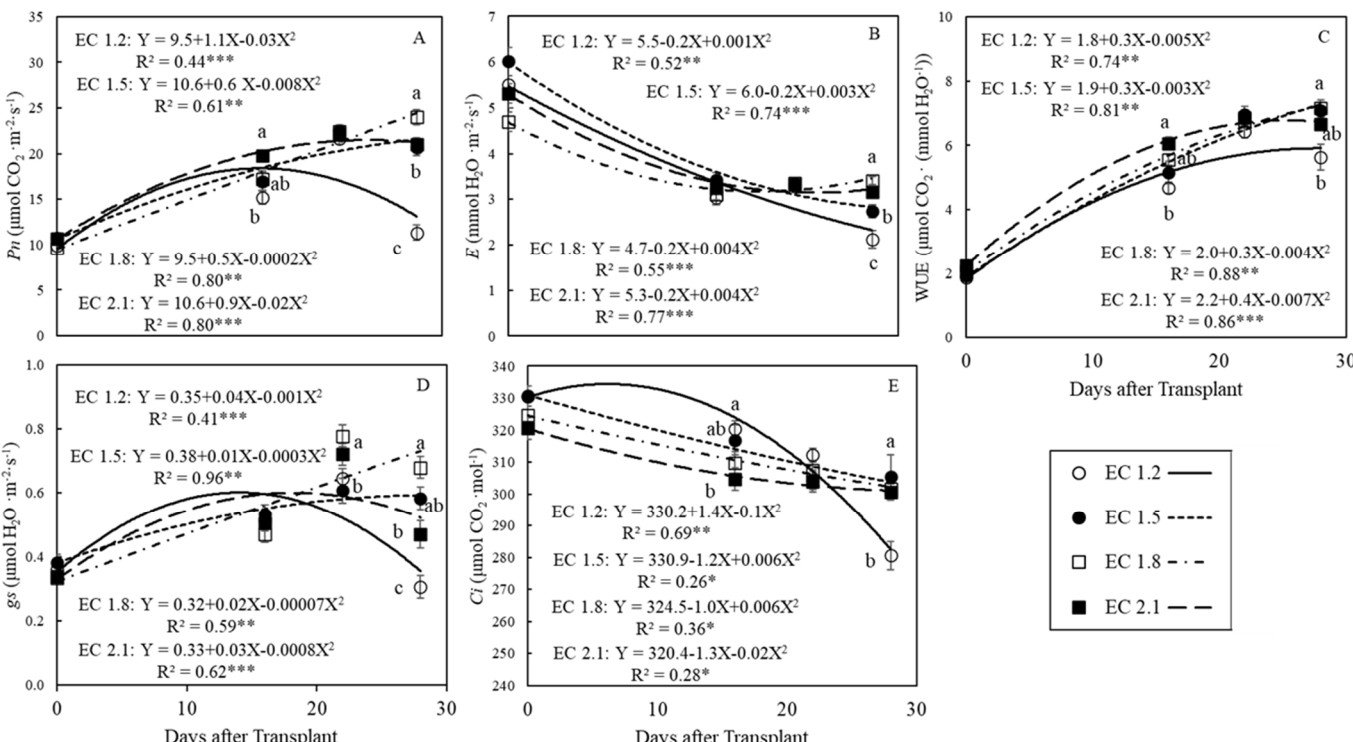

**Figure 6.** Effect of electrical conductivity (EC) of nutrient solution on (**A**) the net photosynthetic rate, (**B**) transpiration rate, (**C**) water use efficiency, (**D**) stomata conductance, and (**E**) intracellular $CO_2$ concentration of arugula on the day before transplant, and 16th, 22nd, and 28th days after transplanting into a nutrient-film technique hydroponic system containing nutrient solutions with EC of 1.2, 1.5, 1.8, or 2.1 dS·m$^{-1}$. Data points with different letters are significantly different according to Tukey's test ($\alpha$ = 0.05). Error bars represent the standard error (*n* = 16). *, ** or *** indicate significance at *p* of 0.05, 0.01 or 0.001, respectively.

The net photosynthetic rate (*Pn*) increased with the higher EC treatment and EC 2.1 dS·m$^{-1}$ had the highest value in the second week. However, there was no significant difference among treatments in the third week and EC 1.8 dS·m$^{-1}$ had the highest value in the final week as a result of decreasing of Pn in other treatments.

The transpiration rate (E) did not show any significant difference in the first three weeks, and initial E was 51% to 82% higher than the following weeks when arugula seedlings were still young after transplant. However, after arugula matured, there was a significant difference among treatment with EC values 1.8 > 2.1 > 1.5 > 1.2 dS·m$^{-1}$.

In the second week, WUE increased with the higher EC treatment. While EC 1.2 dS·m$^{-1}$ had significantly lower value, there was no significant difference among other treatments.

The stomata conductance (*gs*) increased with the higher EC treatment in the mature stage, and EC 1.8 dS·m$^{-1}$ had the highest value in the third and final week. The intracellular $CO_2$ concentration (*Ci*) decreased with the higher EC treatment in the second and third week, but in the final week, *Ci* value was significantly lower in EC 1.2 dS·m$^{-1}$, which was in line with its lower *gs* value.

At the end of the study, light response curves were measured in arugula of different EC levels (Figure 7). Data showed that different EC levels affected light response of arugula. With the increased EC values, the maximum gross photosynthetic rate ($P_{gmax}$), maximum net photosynthetic rate ($P_{max}$), and light saturation point (LSP) increased first then decreased, and EC 1.5 and 1.8 dS·m$^{-1}$ had the highest $P_{gmax}$ and $P_{max}$. In addition, EC 1.8 dS·m$^{-1}$ also had the highest apparent quantum yield (AQE) and lowest respiratory rate ($R_D$), which indicated the best light use efficiency.

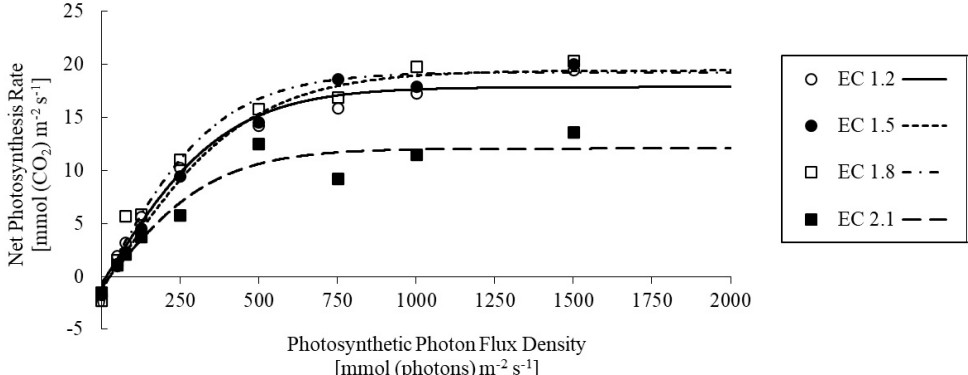

| Treatment | $P_{gmax}$ ($\mu mol \cdot CO_2 \cdot m^{-2} \cdot s^{-1}$) | $P_{max}$ ($\mu mol \cdot CO_2 \cdot m^{-2} \cdot s^{-1}$) | LSP ($\mu mol \cdot m^{-2} \cdot s^{-1}$) | LCP ($\mu mol \cdot m^{-2} \cdot s^{-1}$) | $R_D$ ($\mu mol \cdot m^{-2} \cdot s^{-1}$) | AQE (mol $CO_2 \cdot$ mol photons$^{-1}$) |
|---|---|---|---|---|---|---|
| EC 1.2 | 18.6 | 16.7 | 665 | 14.4 | 0.7 | 0.047 |
| EC 1.5 | 20.6 | 18.1 | 763 | 26.2 | 1.2 | 0.045 |
| EC 1.8 | 20.0 | 18.1 | 652 | 13.0 | 0.7 | 0.054 |
| EC 2.1 | 13.0 | 11.0 | 550 | 25.8 | 0.9 | 0.037 |

**Figure 7.** Effect of electrical conductivity (EC) of nutrient solution on the light response curve, maximum gross photosynthetic rate ($P_{gmax}$), maximum net photosynthetic rate ($P_{max}$), light saturation point (LSP), light compensation point (LCP), respiratory rate ($R_D$), and apparent quantum yield (AQE) of arugula 4 weeks after transplanting into a nutrient-film technique hydroponic system containing nutrient solutions with EC of 1.2, 1.5, 1.8, or 2.1 dS·m$^{-1}$.

### 3.3. Leaf Nutrient and Nitrate Contents

In general, nutrient content increased with the higher EC levels, but leaf tissues in the highest EC, 2.1 dS·m$^{-1}$, showed a significant decline in N, Ca and most of the micronutrients including Fe, Zn, Mo, Cu, B and Mn. While potassium continuously accumulated in arugula leaf with the increase of EC levels (Figure 8).

Nutrient concentration of N (Figure 8A) and Ca (Figure 8D) were highest in EC 1.5 dS·m$^{-1}$, but there was no significant difference between EC 1.5 and EC 1.8 dS·m$^{-1}$. The N concentrations of EC 1.5 and 1.8 dS·m$^{-1}$ were 104 to112% and 28 to 33% higher than EC 1.2 and 2.1 dS·m$^{-1}$, respectively. While the Ca concentrations of EC 1.5 and 1.8 dS·m$^{-1}$ were 6% and 31% higher than EC 1.2 dS·m$^{-1}$. Concentration of K (Figure 8C) was highest in EC 2.1 dS·m$^{-1}$, which may have been caused by the antagonism of N uptake in EC 1.5 and EC1.8 dS·m$^{-1}$. The K concentrations of EC 2.1 dS·m–1 were 91%, 27% and 23% higher than EC 1.2, 1.5 and 1.8 dS·m$^{-1}$, respectively. The macro nutrient concentrations of P (Figure 8B), Mg (Figure 8E) and S (Figure 8F) were not affected by EC treatments.

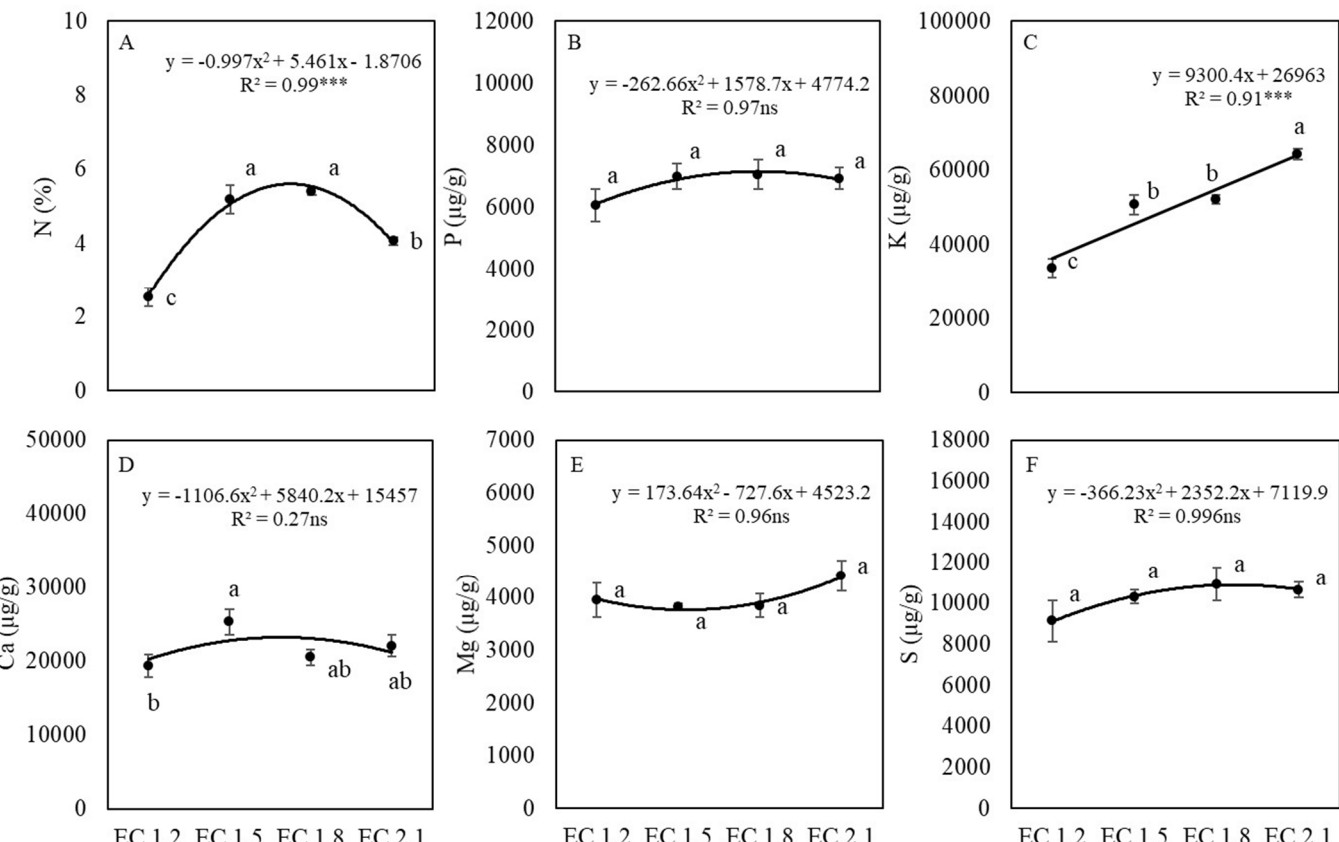

**Figure 8.** Effect of electrical conductivity (EC) of nutrient solution on arugula macro nutrition content including: (**A**) nitrogen, (**B**) phosphorus, (**C**) potassium, (**D**) calcium, (**E**) magnesium, and (**F**) sulfur in the shoot part of arugula 4 weeks after transplanting into a nutrient-film technique hydroponic system containing nutrient solutions with EC of 1.2, 1.5, 1.8, or 2.1 dS·m$^{-1}$. Data points with different letters are significantly different according to Tukey's test ($\alpha$ = 0.05). Error bars represent the standard errors (*n* = 32). ns or *** indicate no significance or significance at *p* of 0.001.

Nutrient concentrations of Fe (Figure 9A), Zn (Figure 9B) and Mo (Figure 9C) were highest in EC 1.5 dS·m$^{-1}$, while that of Cu (Figure 9D), B (Figure 9E) and Mn (Figure 9F) were highest in EC 1.8 dS·m$^{-1}$. Except Mn, there were no significant differences in nutrient concentrations between EC 1.5 and EC 1.8 dS·m$^{-1}$. The concentrations of Fe, Zn, and Mo of EC 1.5 dS·m$^{-1}$ were 36–44%, 41–51% and 101–152% higher than EC 1.2 and 2.1 dS·m$^{-1}$, respectively. The concentrations of Cu and B of EC 1.8 dS·m$^{-1}$ were 112–245% and 34–39% higher than EC 1.2 and 2.1 dS·m$^{-1}$, respectively. The concentrations of Mn of EC 1.8 were 149%, 60% and 155% higher than EC 1.2, 1.5 and 2.1 dS·m$^{-1}$, respectively.

Although there was no significant difference in nitrate concentration between EC 1.8 and EC 2.1 dS·m$^{-1}$, nitrate concentration increased with increased EC (Figure 10). Compared to EC 1.2 dS·m$^{-1}$, nitrate concentration increased by 16 times, 59 times and 75 times in EC 1.5, EC 1.8 and EC 2.1 dS·m$^{-1}$, respectively.

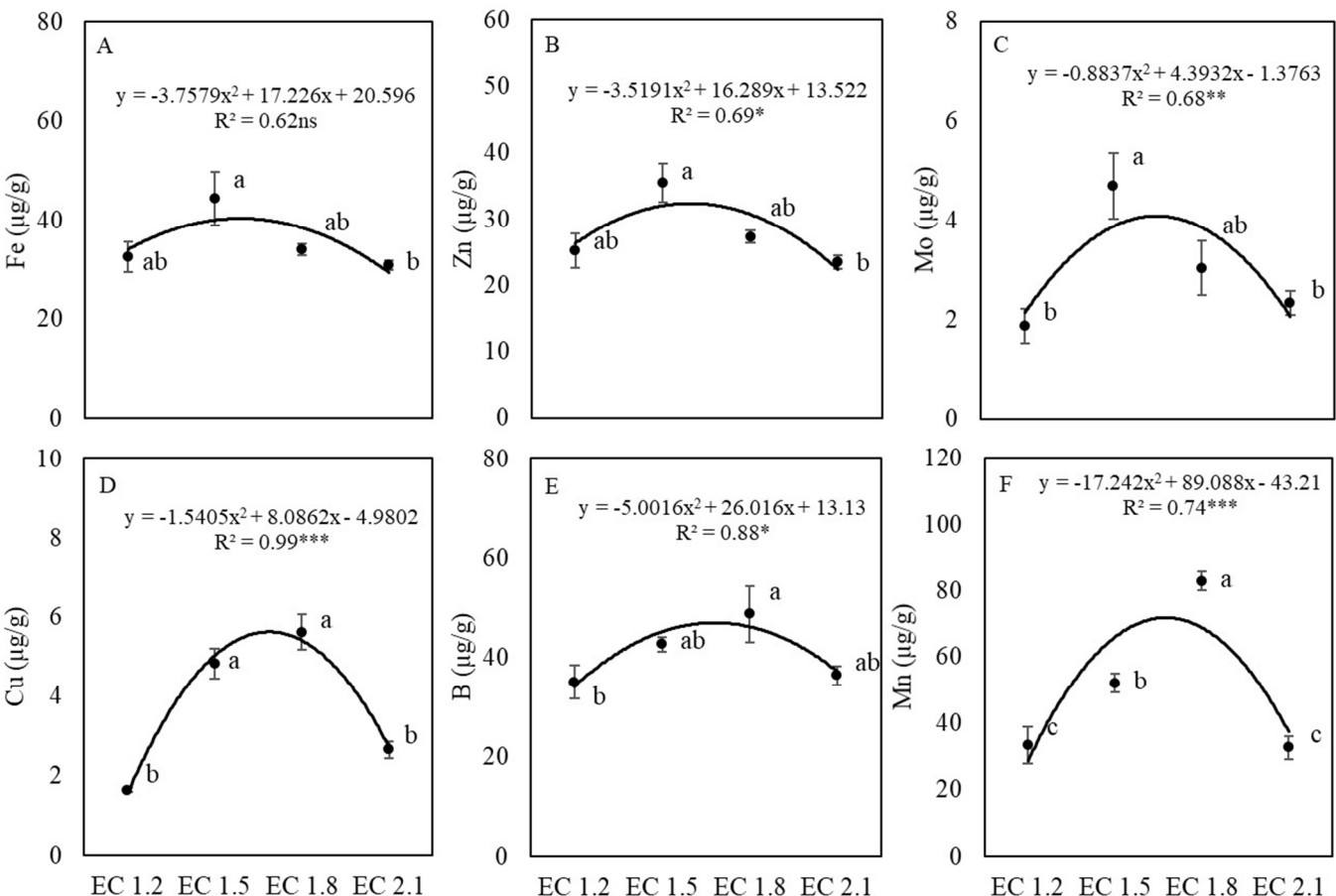

**Figure 9.** Effect of electrical conductivity (EC) of nutrient solution on arugula micro nutrition contents of (**A**) iron, (**B**) zinc, (**C**) molybdenum, (**D**) copper, (**E**) boron and (**F**) manganese in the shoot part of arugula 4 weeks after transplanting into a nutrient-film technique hydroponic system containing nutrient solutions with EC of 1.2, 1.5, 1.8, or 2.1 dS·m$^{-1}$. Data points with different letters are significantly different according to Tukey's test ($\alpha$ = 0.05). Error bars represent the standard errors ($n$ = 32). ns, *, ** or *** indicate no significance or significance at $p$ of 0.05, 0.01 or 0.001, respectively.

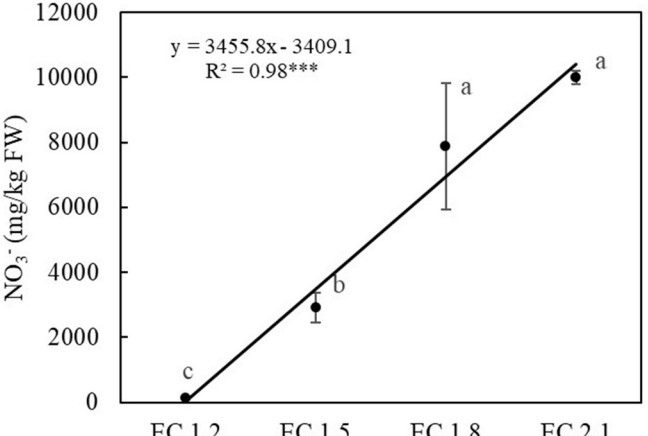

**Figure 10.** Effect of electrical conductivity (EC) of nutrient solution on the nitrate concentration of arugula leaves 4 weeks after transplanting into a nutrient-film technique hydroponic system containing nutrient solutions with EC of 1.2, 1.5, 1.8, or 2.1 dS·m$^{-1}$. Data points with different letters are significantly different according to Tukey's test ($\alpha$ = 0.05). Error bars represent the standard errors ($n$ = 32). *** indicatea significance at $p$ of 0.001.

### 3.4. Leaf Phytochemical Content

Total glucosinolates, total chlorophyll and total carotenoids (Figure 11) increased quadratically with increasing EC which were significantly higher in EC 1.8 dS·m$^{-1}$ and significantly lower in EC 1.2 dS·m$^{-1}$. However, the total anthocyanin had the opposite trends compared to glucosinolates and other phytochemicals. Compared to EC 1.2 dS·m$^{-1}$, the total glucosinolates of EC 1.5, 1.8 and 2.1 dS·m$^{-1}$ were higher by 24%, 41% and 19%, respectively. In terms of total carotenoids, values in EC 1.5, 1.8 and 2.1 dS·m$^{-1}$ were higher by 34%, 50% and 8%, respectively, than that in EC 1.2 dS·m$^{-1}$. And total chlorophyll was enhanced by 34%, 51% and 9%, respectively, in EC 1.5, 1.8 and 2.1 dS·m$^{-1}$ than that in EC 1.2 dS·m$^{-1}$. But there was no significant difference among EC 1.5, 1.8 and 2.1 dS·m$^{-1}$. On the other hand, the total anthocyanin of EC 1.5, 1.8 and 2.1 dS·m$^{-1}$ were decreased by 34%, 50% and 30%, respectively, when compared with that in EC 1.2 dS·m$^{-1}$. But there was no significant difference between EC 1.2 and 2.1 dS·m$^{-1}$, or between EC 1.5 and 1.8 dS·m$^{-1}$.

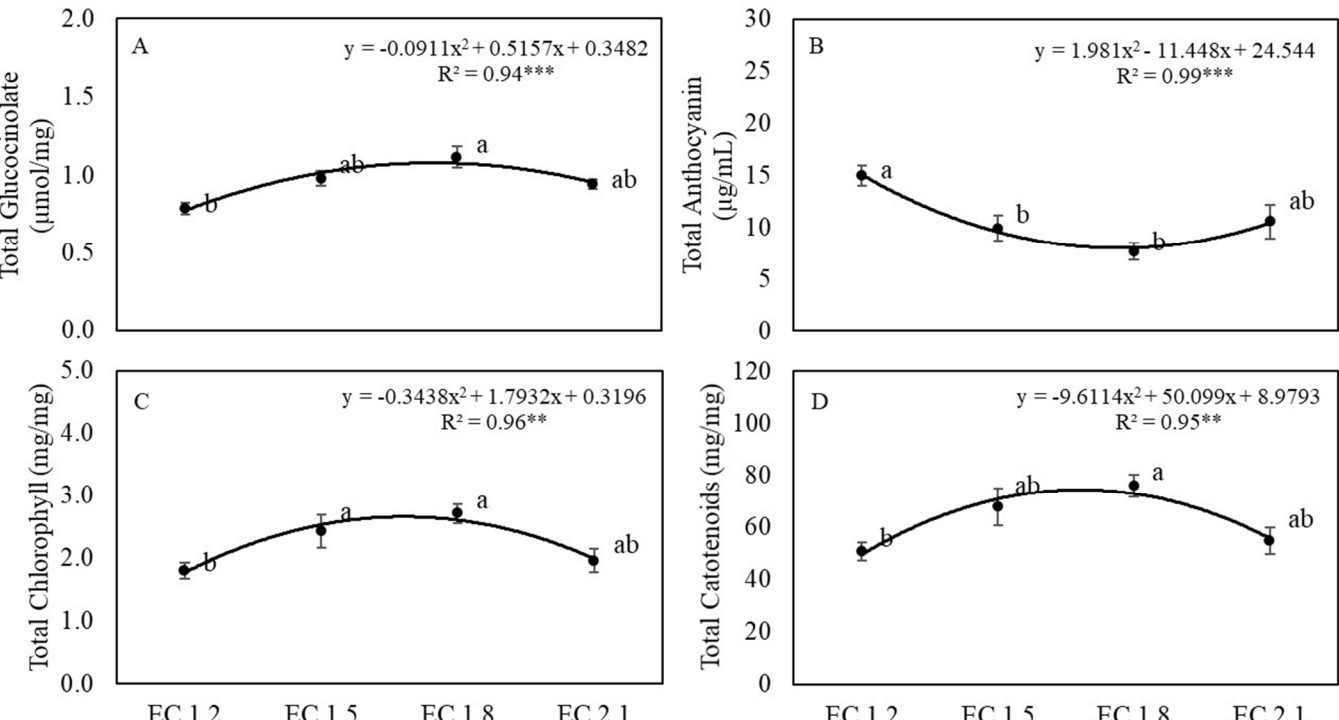

**Figure 11.** Effect of electrical conductivity (EC) of nutrient solution on the arugula phytochemical content of (**A**) total glucosinolates, (**B**) total anthocyanin, (**C**) total chlorophyll, and (**D**) carotenoids in the shoot part of arugula 4 weeks after transplanting into a nutrient-film technique hydroponic system containing nutrient solutions with EC of 1.2, 1.5, 1.8, or 2.1 dS·m$^{-1}$. Data points with different letters are significantly different according to Tukey's test ($\alpha$ = 0.05). Error bars represent the standard errors (*n* = 32). ** or *** indicate significance at *p* of 0.01 or 0.001.

## 4. Discussion

### 4.1. EC Affected Growth and Yield of Arugula

The shoot fresh and dry yields in our study were 17 to 42 g fresh weight per plant and 2.6 to 4.2 g dry weight per plant among EC treatments. These values were comparable with Kim et al. [35] and more than three times higher than Santamari et al. [24] and Bonasia et al. [3] in hydroponic-grown arugula production experiments. As expected, EC 1.2 dS·m$^{-1}$ showed the lowest leaf area and SPAD value due to insufficient nutrition. There was an obvious tendency of greater leaf area with the increase of the EC, but the SPAD, marketable yield and shoot:root ratio were slightly reduced in EC 2.1 dS·m$^{-1}$. In addition, EC 1.2 and 2.1 dS·m$^{-1}$ had very severe nutrient disorder symptoms. All these results indicated that arugula growth and yield were negatively affected by too high or too low EC level (nutrient concentration), which had been established by previous research

in hydroponics production with other leafy vegetables including lettuce [36–41], pakchoi (*Brassica rapa*) [42], table beets (*Beta vulgaris*) [36], and watercress (*Nasturtium officinale*) [43], but none of these research investigated the response to EC for arugula.

In the current study, arugula growth and yield parameters all increased when EC increased from 1.2 to 1.8 dS·m$^{-1}$, then decreased when EC further increased to 2.1 dS·m$^{-1}$ which may be due to high osmotic pressure [12,44,45]. It had been found that both nutrient deficiency (too low EC) and osmotic stress (too high EC) could lead to a reduction in chlorophyll content [37,46], yields [36], imbalance between shoot and root growth [37], and lead to symptoms typical of nutritional disorders [12]. Although the response of most leafy vegetables to increasing EC fits a quadratic polynomial model, the response degree/sensitivity differs greatly with species. The optimum EC levels were EC 1.5 and 1.8 dS·m$^{-1}$ in our study, with total nitrogen application around 110 to 150 mg·L$^{-1}$. Murphy & Pill [47] also found that providing a daily fertilizer solution with 150 mg·L$^{-1}$ N could increase shoot fresh weight·m$^{-2}$ for microgreen arugula grown in perlite media. In this study, the shoot:root ratio increased when EC increased from 1.2 to 1.8 dS·m$^{-1}$, then decreased when EC reached 2.1 dS·m$^{-1}$, which was likely due to similar root biomass and concomitant reduction of shoot mass imposed by the osmotic stress. However, Campos et al. [48] used brackish water in NFT hydroponic arugula production under EC levels of 1.5, 3.0, 4.5, 6.0, 7.5, and 9.0 dS·m$^{-1}$, and found a linear increase of root biomass and decrease of shoot:root ratio with the increase of EC in saline nutrient solution which were not consistent with the results of our study. Thus, the different responses of arugula to EC levels in our study could be due to the nutrient supply and high osmotic potential in the study of Campos et al. [48].

### 4.2. EC Affected Leaf Nutrient Concentrations of Arugula

In order to evaluate the nutrient contents in leaf tissue, a comparison was conducted according to the sufficiency ranges for arugula [49]. In all treatments, tissue nutrient concentrations were generally within or above sufficiency ranges, except N, K, Ca, Fe, Mn, Cu, Zn and Mo. It was found that macronutrient N content was less than the sufficiency range under EC of 1.2 and 2.1 dS·m$^{-1}$, which could explain the reduction of chlorophyll content (SPAD) in these two treatments. What is more, macronutrient K content was also less than the sufficiency range under EC of 1.2 dS·m$^{-1}$. In addition, it was also found that micronutrient Mn and Cu content was below the sufficiency range under EC of 1.2 and 2.1 dS·m$^{-1}$, while Ca content was below the range under EC of 1.2 dS·m$^{-1}$. We contributed the lack of N, K and Cu under EC of 1.2 dS·m$^{-1}$ to the sub-optimal nutrient supply in this treatment, which also could be reflected by the 45% to 53% reduction of Pn in EC 1.2 dS·m$^{-1}$ compared with other treatments during the study. These results were in agreement with previous research that investigated nutrient concentration effects using basil (*Ocimum basilicum*) [50] and chicory (*Chicorium Spinosum*) [51]. Chatzigianni et al. [51] found total-N concentration did not show any consistent impact on the growth and yield of hydroponic grown chicory.

Surprisingly, Fe, Mo and Zn contents of arugula in our study were less than the sufficiency range, even in the optimum EC treatments (EC 1.5 and 1.8 dS·m$^{-1}$) with highest yield. These results may be caused by the pH spike during the study (Table 1). Dickson & Fisher [52] also found arugula had highest root zone pH spike than any other vegetables in their hydroponic research, which indicated a possibility of nutritional problem [53]. Thus, with the application of EC 1.5 to 1.8 dS·m$^{-1}$, we also emphasize the importance of the appropriate pH monitoring in hydroponic arugula production to further improve the arugula nutritional quality. In addition, we attribute the lack of N and Cu under EC of 2.1 dS·m$^{-1}$ to the osmotic stress, which was also reflected by the 19% to 31% reduction of gs and the 7% to 8% reduction of SPAD in EC 2.1 dS·m$^{-1}$ compared with EC 1.5 and 1.8 dS·m$^{-1}$. These finding were in line with previous research which reported the photosynthetic pigment damage [54] and the reduction of gs [55] under long term mild osmotic stress.

Similar to yield data, most macro and micronutrient concentrations also a fit quadratic polynomial model with increasing EC, except K concentration which fit a linear model with a positive slope. This finding was similar with our previous research of K uptake in NFT-grown lettuce [39,40]. Potassium is the second most absorbed nutrient ion taken up by plants [56]. However, potassium can be actively absorbed by roots in a few hours, so this essential nutrient may lead to toxic accumulation in plant tissue [57]. According to arugula nutrient sufficiency range [49], K concentration of arugula shoots under EC 2.1 dS·m$^{-1}$ in our study was higher than the upper level of the sufficiency range which may suggest a potential toxicity. Although nutrition management through EC is preferable by growers in commercial hydroponic production, and hydroponic fertilizers are formulated to balance nutrient composition with known plant uptake rates, this management practice could lead to nutrient imbalances during production if one or more nutrients are taken up more quickly than others.

### 4.3. EC Affected Leaf Gas Exchange Indices in Arugula

Electrical conductivity of the nutrient solution is directly related to the photosynthetic metabolism [34]. In our study, except EC 1.2 dS·m$^{-1}$, most of the leaf gas exchange indices did not show significant difference among other treatments in the first two or three weeks. After the fourth week, net photosynthetic rate, transpiration rate and stomatal conductance showed a curvilinear pattern in response to the applied nutrient levels. These indices increased gradually with increasing EC, reached their maximum levels between 1.5 and 1.8 dS·m$^{-1}$, then declined with further EC increases to 2.1 dS·m$^{-1}$, which could be a result of high osmotic potential in the nutrition solution [58,59]. A similar trend was also reported by Ding et al. [42] and Tabatabaie & Nazari [45] who investigated the optimized EC for hydroponic-grown pakchoi and lemon verbena, respectively. Albornoz & Lieth [58] interpreted the yield reduction of lettuce with higher nutrient concentration as a combined effect of decreased stomatal conductance and leaf area. However, in our study, leaf area continued to increase with increasing EC levels, so stomatal conductance was the critical factor for arugula growth in this study. Higher stomatal conductance is critical for plant growth because it could enhance $CO_2$ supply, thus improve net photosynthetic rate [34], which could further explain the higher yields in EC 1.5 and 1.8 dS·m$^{-1}$.

The light response curve results indicated the better light use efficiency in EC 1.5 and 1.8 dS·m$^{-1}$, which agree with the leaf nitrogen content results. The plant photosynthetic capacity was highly correlated with N content of the leaves [60,61], because N is the basic constituent of proteins involved in photosynthetic activity. And the lower photosynthetic rate could explain the lower assimilate production in EC 1.2 and EC 2.1 dS·m$^{-1}$, which was similar with previous research [39,62–64].

### 4.4. EC Affected Plant Phytochemical Content

Leaf phytochemical contents in arugula increased quadratically with increasing EC levels, and the opposite influences were observed in total anthocyanin and other phytochemicals (total glucosinolates, total chlorophyll and total carotenoids). Optimized EC (1.5 and 1.8 dS·m$^{-1}$) for maximum yield also promoted antioxidants compounds except anthocyanin. Chen et al. [65] also indicated that application of medium N concentration (50 to 100 mg/L) in nutrient solution significantly enhanced chlorophyll and carotenoids accumulation in Chinese kale (Brassica alboglabra Bailley), which was similar to results of Fallovo et al. [37] with lettuce. However, under environmental stresses, total chlorophyll has been reported to decrease as total anthocyanin content increased [66,67], which was in agreement with our study. Chlorophyll breakdown is a process related with leaf senescence which could happen when plants grow under suboptimal nutrient levels. Low to moderate nitrogen levels have been shown to stimulate the formation of enzymes involved with anthocyanin synthesis [68–70]. Both Gershenzon [71] and Akula & Ravishanka [72] attributed the formation of anthocyanins to nutrient stress in general.

Although some research reported no effect [37,73] or negative effect [74] of nutrient levels on carotenoids formation in lettuce, Vernieri et al. [75] and Alberici et al. [76] observed an increase of total chlorophyll and carotenoids in arugula with increasing nutrient concentrations, which are similar to findings in this investigation. This could occur due to the formation of photosynthetic organelle as a result of chlorophyll accumulation.

Glucosinolates and other plant-derived antioxidant compounds play a role in abiotic and biotic stress with a process of free-radical scavenge [77]. It was found that nutrient levels had a regulated effect on the biosynthesis and catabolism of plant glucosinolates [2], thus an optimized nutrition was needed to maximize glucosinolates contents. In our study, only EC of 1.2 dS·m$^{-1}$ showed a significant reduction in glucosinolates accumulation, and EC of 2.1 dS·m$^{-1}$ showed a slight reduction although no significant difference than EC of 1.8 dS·m$^{-1}$. Glucosinolates contribute to pungent or bitter flavor in the family Brassicaceae [78–80], while anthocyanin and carotenoids contribute to coloration [81,82]. In addition, Schonhof et al. [80] found that consumers prefer arugula cultivars with less bitter/pungent glucosinolates and color intensity. Thus, considering the health benefits, flavor functions as well as the consumer preference of glucosinolates, anthocyanin and carotenoids, EC of 1.5 to1.8 dS·m$^{-1}$ would be best EC level for hydroponic grown arugula.

*4.5. EC Affected Leaf Nitrate Content and Potential Influence on Human Health*

The current study showed a steep rise in arugula leaf nitrate content with increasing EC. High amount of dietary nitrate may have negative effects on human health due to the 'nitrate-nitrite-nitric oxide (NO) pathway', which includes an association with gastric and bladder cancers [83,84], and a nitrite implication in the methaemoglobinaemia syndrome [85]. It has been reported that arugula is a hyper-accumulator of nitrates which could be as high as 7000 to 8000 mg·kg$^{-1}$ [23,24], and hydroponic culture would further enhance nitrate accumulation [86,87]. Thus, factors such as effect of fertilization rate on nitrate accumulation should be taken into consideration in hydroponics.

Despite contradictory opinions about the negative effect of nitrate content on cancer incidence [85,88,89], considering the high cancer risk of certain subgroups of the human population, the World Health Organization (WHO) indicates a daily nitrate intake threshold of 3.7 mg·kg$^{-1}$ body weight per day [90]. In addition, Europe established related regulations to limit the nitrate content in vegetables [2]. According to Regulation (EU) No. 1258/2011, nitrate content in commercial arugula should be below 6000 mg·kg$^{-1}$ FW in summer production cycle or 7000 mg·kg–1 FW in winter production cycle [91]. In the current study conducted in winter, we found arugula grown under EC 1.8 and 2.1 dS·m$^{-1}$ had the nitrate concentration exceeded the threshold (7000 mg·kg–1 FW) by 12.5% and 42.9%, while that in EC 1.2 and 1.5 dS·m$^{-1}$ were within the acceptable range. Therefore, with the concern of human health, EC application for hydroponic-grown arugula should be lower than EC 1.8 dS·m$^{-1}$.

## 5. Conclusions

Based on yield, quality criteria and human health, the optimal EC treatment would be EC 1.5 to 1.8 dS·m$^{-1}$ for arugula (*Eruca sativa* Mill. 'Standard') in the NFT hydroponic production system. Too low and too high EC would reduce yields, visual quality, phytochemical compounds and lead to a less attractive color and taste to consumer and enhance the negative health effects due to nitrate accumulation.

**Author Contributions:** Conceptualization, U.S. and T.Y.; methodology, T.Y., U.S., J.A. and P.L.; formal Analysis, T.Y. and U.S.; investigation, T.Y.; writing—original draft preparation, T.Y.; writing—review and editing, U.S., J.A. and P.L.; supervision, U.S. and J.A.; project administration, U.S.; funding acquisition, U.S. and J.A. All authors have read and agreed to the published version of the manuscript.

**Funding:** This research was funded by United States Department of Agriculture—Agricultural Research Service, grant number GRT00056364.

**Data Availability Statement:** The data presented in this study are available on request from the corresponding author.

**Acknowledgments:** The authors would like to thank Wee Fong Lee for his technical assistance; Sean Fitzgerald, Makayla Miller for their help with system preparation, crop management and data collection; Leslie Morris and Dee Marty for their help with data collection.

**Conflicts of Interest:** The authors declare no conflict of interest.

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
