# Peer review of "Photosynthesis, Biomass Production, Nutritional Quality, and Flavor-Related Phytochemical Properties of Hydroponic-Grown Arugula (Eruca sativa Mill.) ‘Standard’ under Different Electrical Conductivities of Nutrient Solution"

_agronomy, doi:10.3390/agronomy11071340_

Round 1
Reviewer 1 Report
The aim of this study is to evaluate morphological, nutritional and physiological properties of arugula grown in hydroponics with different electrical conductivities. The idea of research is interesting, study brings new knowledge, and may have practical value for greenhouse growers. In my opinion, research work was performed good, however I have some suggestions:
I suggest including cultivar name into manuscript title, study was conducted with 1 cultivar, therefore its too optimistic to make conclusions for whole species based on such data. Also, the same should be made thorough manuscript, including conclusions.
There is paragraph from template left in section 3. Results.
Some figures lack resolution. Also, I would suggest making figures in colour.
Figure 6 lacks statistical evaluation.
In general, I like the idea and presentation of the manuscript, I suggest publishing after minor revision.
Reviewer 2 Report
This is an interesting and well written paper about effect of EC on growth and nutritional properties of arugula.
However, I have a few comments which should be addressed: First, you can consider adding in the title the scientific name of the plant (Eruca sativa Mill.). Next, I have a few comment to the methodology:
You used Hydro Grow Leafy Green Fertilizer; 4.3% N-9.3%P-3.5%K, please check if K concentration is correct. I think it should be 35%. I think also that this fertilizer is not a ready prepared solution. You used 100 mL/L. Is it about fertilizer itself or water stock solution, if so, how was it prepared? The same about calcium nitrate. Please specify.
In Table 2 the concentration of nitrate in nutrient solution (557.8 mg/L) seems to be very high. Maybe you mistakenly mixed it with the amount of added calcium nitrate (15.5% N)? If so, please check also another parameters presented in the table.
Reviewer 3 Report
Please find specific place of comments by attached file .
- Fig. S1: I could not this fig., Please keep the right format
- Line 142: Add (TOC)
- Line 148: Where is table 1?
- Table 1: This table has no headline and the right side is not visible
- Lines 274-275: why did you measure it only at one day? only 28?
- Figure 1: Please mention the day of measurement here. At which day?
- Line 310: replace it with: (F) shoot to root weight ratio
- Fig. S2: It would be better if you could explain disorder symptom and compare it in different EC treatments.
- Fig. 5: You have both letters and error bars. mention letters as well.
- Line 356: Is this table or fig?
- Fig. 7: You have both error bars and letters. Please mention the letters as well. Apply it in all figures.
- Lines 521-522: It seems that line spacing is smaller than journal format. Is it right? Please check it once more through the entire manuscript.

Reviewer 4 Report
Main comments
The Authors took up an interesting topic related to the production of vegetables in a hydroponic system and the problem of concentration of the nutrient solution and its associated electrical conductivity. These are very useful and difficult to conduct research.
The research undertaken is in line with the current global trends of seeking solutions to save water, fertilizers and energy and the search for locally produced food of the highest quality. The work fits in with the research subject of Agronomy.
The topic of the manuscript is very relevant and the manuscript is well-written.
The methodological assumptions and execution of the experiment are accurate and detailed. Clearly and precisely presented results and skillfully initiated discussion. The work is interesting and well prepared.
Comments and suggestions:
Line 4: I would suggest adding in the title:…under different electrical conductivities of nutrient solution
Line 26: Keywords: Eruca sativa (italic font)
Line 110: Figure S1 (the figure 1 was not fully visible in the received version for pdf review)
Line 113-114: What kind of high intensity discharge lamps (HID) was used in the experiments (name, power, producer) whether they were high-pressure sodium lamps (HPS) or fluorescent or LED lamps? Is artificial lighting was used together with natural light or separately?
Line 120: using Arugula (Eruca sativa) cultivar
Line 118-119: should be: daily light integral or daily light intensity readings of PPFD. (This is indicated by the values: 17.03 ± 1.71 mmol·m–2·d–1)
Line 150: should be: Table 1. (the table 1 was not fully visible in the received version for pdf review).
Line 305: nutrient excess causing toxicity to plants
Line 468-469: basil (Ocimum basilicum) and chicory (Chicorium Spinosum).
